# Safe Bayesian Optimization for the Control of High-Dimensional Embodied Systems

**Yunyue Wei, Zeji Yi, Hongda Li, Saraswati Soedarmadji, Yanan Sui**
Tsinghua University
`weiyy20@mails.tsinghua.edu.cn`
`zejiy@andrew.cmu.edu`
`lhd21@mails.tsinghua.edu.cn`
`chenxuying24@mails.tsinghua.edu.cn`
`ysui@tsinghua.edu.cn`

**Abstract:** Learning to move is a primary goal for animals and robots, where ensuring safety is often important when optimizing control policies on the embodied systems. For complex tasks such as the control of human or humanoid control, the high-dimensional parameter space adds complexity to the safe optimization effort. Current safe exploration algorithms exhibit inefficiency and may even become infeasible with large high-dimensional input spaces. Furthermore, existing high-dimensional constrained optimization methods neglect safety in the search process. In this paper, we propose High-dimensional Safe Bayesian Optimization with local optimistic exploration (HDSAFEBO), a novel approach designed to handle high-dimensional sampling problems under probabilistic safety constraints. We introduce a local optimistic strategy to efficiently and safely optimize the objective function, providing a probabilistic safety guarantee and a cumulative safety violation bound. Through the use of isometric embedding, HDSAFEBO addresses problems ranging from a few hundred to several thousand dimensions while maintaining safety guarantees. To our knowledge, HDSAFEBO is the first algorithm capable of optimizing the control of high-dimensional musculoskeletal systems with high safety probability. We also demonstrate the real-world applicability of HDSAFEBO through its use in the safe online optimization of neural stimulation induced human motion control.

**Keywords:** Safe Bayesian Optimization, High-dimensional Embodied System

## 1 Introduction

Some robotics applications require online optimization of control policies for performance while avoiding unsafe parameter tuning that could potentially damage the systems. These scenarios correspond to the problem of *safe exploration*, which involves the sequential optimization of an unknown objective function under the constraint of satisfying unknown safety conditions. Bayesian optimization (BO) is an effective paradigm in optimizing black-box functions, and safe BO methods have been successfully used to tune control parameters for various robotic systems [1, 2, 3]. Most existing safe BO methods use the Gaussian process (GP) to model the underlying safety function, and discriminate safe regions with estimated function's lower confidence bound to ensure safety with high probability [4]. Such conservative strategies are inefficient for objective optimization, and even infeasible in high-dimensional input settings, such as human or humanoid system control.

A motivating application of our work is the control of musculoskeletal (tendon-driven) systems, where complex motions are driven by dozens to hundreds of muscle-tendon units rather than joints. Such overactuated embodied systems introduce additional control challenges within a large-scale parameter space. Under this high-dimensional input setting, efficiently optimizing the task function while

8th Conference on Robot Learning (CoRL 2024), Munich, Germany.

maintaining safety remains a formidable challenge for existing safe optimization algorithms. Despite considerable efforts to leverage BO for solving high-dimensional constrained optimization problems, these methods often fail to incorporate safety considerations into the optimization procedure [5, 6, 7]. To our best knowledge, there are currently no methods that guarantee safety, or probabilistic safety during the high-dimensional optimization.

In this paper, we introduce High-dimensional Safe Bayesian Optimization with local optimistic exploration (HDSAFEBO) to address probabilistic safety in high-dimensional sequential optimization problems, as shown in Figure 1. To achieve efficient and safe optimization, we propose a *local optimistic strategy* with probabilistic safety guarantee and cumulative safety violation bound. By using isometric embedding for dimension reduction, we enable HDSAFEBO to handle even higher dimensional problems, ranging from a few hundred to several thousand dimensions, while maintaining the probabilistic safety guarantee. Our experimental results show that HDSAFEBO efficiently learns to control a musculoskeletal system with high safety probability - a task where all baseline methods fail. We also demonstrate the success of HDSAFEBO in real-world experiments to safely optimize the control of neural stimulation induced human motion. Our project page is at `https://lnsgroup.cc/research/hdsafebo`

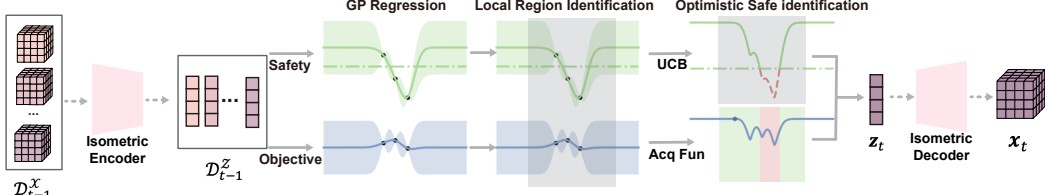

Figure 1: **Workflow of** HDSAFEBO. Local optimistic safe optimization is employed to efficiently optimize the objective function while guaranteeing probabilistic safety. We utilize isometric embedding to reduce the problem dimension to deal with high-dimensional inputs.

## 2 Related Work

### 2.1 Safe Bayesian Optimization

The sequential decision-making problem with safety constraints has been extensively studied, varied by the definition of safety. To achieve full safety during exploratory sampling, algorithms have been proposed with theoretical guarantee in near-optimality and safety with high probability [4, 8, 1, 9, 10]. These methods have been applied in safe parameter tuning of quadrotor [11], robot arm [2] and quadruped robots [3]. These safe optimization algorithms conservatively estimate and expand the safe region, leading to inefficient optimization performance.

In contrast to a zero-tolerance approach to unsafe actions, an alternative approach allows for limited constraint violations within a predefined budget, trading safety for more efficient optimization[12, 13, 14]. A recent work CONFIG uses upper confidence bound to optimistically estimate the safe region, enjoying global optimal guarantee as unconstrained methods [15].

Another extreme case, called constrained Bayesian optimization, has also been used for finding safe controller parameters [16, 17, 18, 19, 20]. However, this line of methods aim only to find the best feasible solution, neglecting the safety during the optimization process. Constrained expected improvement (cEI) is a popular constrained BO algorithm that introduces feasibility constraints to acquisition function formulation [21, 16].

All of the aforementioned methods fall in the framework of Bayesian optimization, which is typically limited to low-dimensional problems[22, 23]. LineBO demonstrates success in optimizing problems with dimension up to 40 via searching over 1-dimensional subspace at each iteration [24, 25]. When parameter size exceeds 10, safe optimization algorithms might even struggle to expand the safe region due to sparse discretization of the input space.

## 2.2 High-dimensional Bayesian Optimization

Recent research efforts have delved into the utilization of Bayesian optimization in high-dimensional problems [26, 27]. In the main paper, our focus lies on dimension reduction-based and local BO methods. Further discussion about other relevant works can be found in Appendix D.

A large body of literature leverages dimension reduction to apply BO over a low-dimensional subspace. Several works use variable selection to identify and optimize important dimensions during optimization [28, 29, 30]. Another popular approach for reducing the search space is random linear embedding, which has been proven to contain the optimal solution with a certain probability relative to the objective function's effective dimension[31, 32, 33, 34, 35]. Many works also use autoencoder to learn a non-linear mapping between the original space and the latent space [36, 37, 38, 39, 40, 41, 6].

Another line of works utilizes local search to address the over-exploration issue in high-dimensional optimization [42], achieving better empirical performance than global BO methods [43, 44, 45, 46]. This local optimization strategy is also able to be combined with dimension reduction [47] and extended to constrained setting (SCBO [7]). Local BO methods have also been deployed for optimizing locomotion control in simulation [48, 30].

Besides BO methods, evolutionary algorithms such as CMA-ES are competitive to solve high-dimensional problems [49] and can be extended to constrained setting[50, 51]. Although many works attempt to solve high-dimensional constrained optimization problems, to the best of our knowledge, there lacks work that addresses safety in high-dimensional sequential optimization with at least hundreds of variables.

## 3 Problem Formulation

We aim to optimize an unknown objective function $f : \mathcal{X} \to \mathbb{R}$ by sequentially sampling points $\boldsymbol{x}_1, \ldots, \boldsymbol{x}_t \in \mathcal{X}$. We can also get observations of safety measurement from another unknown function $g : \mathcal{X} \to \mathbb{R}$. We define a point $\boldsymbol{x}$ is safe when $g(\boldsymbol{x}) > 0$. We can formally write our optimization problem as follows:

$$\max_{\boldsymbol{x}_t \in \mathcal{X}} f(\boldsymbol{x}_t) \quad \text{subject to } g(\boldsymbol{x}_t) \geq 0, \forall t \geq 1, \tag{1}$$

We consider the above problem formulation widely exists in the application of robotic control, where $\boldsymbol{x}$ represents a parameterized controller, and $f(\boldsymbol{x}), g(\boldsymbol{x})$ are the utility and safety measurement of the controller in a single experimental trial. In our target applications such as the control of embodied systems, the input space $\mathcal{X} \in \mathbb{R}^D$ is high-dimensional, encompassing dozens to thousands of variables.

In Bayesian optimization, Gaussian process is usually used as the surrogate model to learn the unknown functions. Taking $g$ as an example, for samples at points $X_t = [\boldsymbol{x}_1...\boldsymbol{x}_t]^T$, we have noise-perturbed observations $\boldsymbol{y}_t = [y_1^g...y_t^g]^T$. The GP posterior over $g$ is also Gaussian with mean $\mu_t(\boldsymbol{x})$, covariance $k_t(\boldsymbol{x}, \boldsymbol{x}')$ and variance $\sigma_t^2(\boldsymbol{x}, \boldsymbol{x}')$ under kernel function $k$:

$$\begin{aligned} \mu_t(\boldsymbol{x}) &= \boldsymbol{k}_t(\boldsymbol{x})^T (\boldsymbol{K}_t + \sigma^2 \boldsymbol{I})^{-1} \boldsymbol{y}_t \\ k_t(\boldsymbol{x}, \boldsymbol{x}') &= k(\boldsymbol{x}, \boldsymbol{x}') - \boldsymbol{k}_t(\boldsymbol{x})^T (\boldsymbol{K}_t + \sigma^2 \boldsymbol{I})^{-1} \mathbf{k}_t(\boldsymbol{x}') \\ \sigma_t^2(\boldsymbol{x}) &= k_t(\boldsymbol{x}, \boldsymbol{x}), \end{aligned} \tag{2}$$

where $\boldsymbol{k}_t(\boldsymbol{x}) = [k(\boldsymbol{x}_1, \boldsymbol{x}), ..., k(\boldsymbol{x}_t, \boldsymbol{x})]$ is the covariance between $\boldsymbol{x}$ and sampled points, $\boldsymbol{K}_t$ is the covariance of sampled positions: $[k(\boldsymbol{x}, \boldsymbol{x}')]_{\boldsymbol{x}, \boldsymbol{x}' \in X_t}$. Similarly we can use GP to derive posterior of $f$. Using the posterior of GP, we can define define $u_t(\boldsymbol{x}) := \mu_{t-1}(\boldsymbol{x}) + \beta_t \sigma_{t-1}(\boldsymbol{x})$ as the upper confidence bound (UCB) of the function estimation, where $\beta_t$ is a scalar which can be properly set to contain $g(\boldsymbol{x})$ with desired probability. we make the following regularity assumptions that are commonly used in the field of Bayesian optimization:

**Assumption 3.1.** $g$ and $f$ are samples of two Gaussian processes defined by the kernels $k(\cdot, \cdot), k'(\cdot, \cdot)$ respectively. The observations are perturbed by i.i.d. Gaussian noise: $y^g(\boldsymbol{x}_t) = g(\boldsymbol{x}_t) + n_t$, $y^f(\boldsymbol{x}_t) = f(\boldsymbol{x}_t) + n'_t$ where $n_t \sim \mathcal{N}(0, \sigma^2), n'_t \sim \mathcal{N}(0, \sigma'^2)$

| **Algorithm 1** HDSAFEBO | **Algorithm 2** SAFEUPDATE |
|---|---|
| **Input** Sample set $\mathcal{X}$, GP priors $GP^f, GP^g$, safety threshold $h$, acquisition function $A$, initial dataset $\mathcal{D}_0^{\mathcal{X}}$, trust region length $l_0$, isometric encoder $\Pi$ and decoder $\Pi^{-1}$ | **Input** Current sample $\boldsymbol{x}_t, y_t^f$, current best sample $\boldsymbol{x}^*, y^*$, current counters $c_s, c_t$ current trust region length $l_t$, success tolerance $\tau_s$, failure tolerance $\tau_f$, initial trust region length $l_0$, trust region limits $l_{\max}, l_{\min}$ |

**Algorithm 1** HDSAFEBO

**Input** Sample set $\mathcal{X}$, GP priors $GP^f, GP^g$, safety threshold $h$, acquisition function $A$, initial dataset $\mathcal{D}_0^{\mathcal{X}}$, trust region length $l_0$, isometric encoder $\Pi$ and decoder $\Pi^{-1}$

1: $c_s, c_f \leftarrow 0$
2: $\boldsymbol{x}^*, y* \leftarrow \text{argmax}_{\{\boldsymbol{x}, y^f\} \in \mathcal{D}_0, y^g(\boldsymbol{x}) > 0} y^f(\boldsymbol{x})$
3: **for** $t = 1$ to $\dots$ **do**
4: $\quad D_{t-1}^{\mathcal{Z}} \leftarrow \Pi(D_{t-1}^{\mathcal{X}})$
5: $\quad$ Update $GP^f, GP^g$ using $\mathcal{D}_{t-1}^{\mathcal{Z}}$
6: $\quad L_t \leftarrow \{\boldsymbol{z} \in \mathcal{Z} \mid \Pi(\boldsymbol{x}^*) - l_t \leq \boldsymbol{z} \leq \Pi(\boldsymbol{x}^*) + l_t\}$
7: $\quad S_t \leftarrow \{\boldsymbol{z}' \in L_t \mid \mu_{t-1}(\boldsymbol{z}) + \beta_t \sigma_{t-1}(\boldsymbol{z}) \geq 0\}$
8: $\quad \boldsymbol{z}_t \leftarrow \text{argmax}_{\boldsymbol{z} \in S_t}(A(\boldsymbol{z}))$
9: $\quad \boldsymbol{x}_t \leftarrow \Pi^{-1}(\boldsymbol{z}_t)$
10: $\quad y_t^f \leftarrow f(\boldsymbol{x}_t) + n_t$
11: $\quad y_t^g \leftarrow g(\boldsymbol{x}_t) + n_t$
12: $\quad \mathcal{D}_t \leftarrow \mathcal{D}_{t-1} \bigcup \{\boldsymbol{x}_t, y_t^f, y_t^g\}$
13: $\quad \boldsymbol{x}^*, y^*, c_s, c_f, l_t \leftarrow$ SAFEUPDATE$(\boldsymbol{x}_t, y_t^f, \boldsymbol{x}^*, y^*, c_s, c_f, l_t)$
14: **end for**

**Algorithm 2** SAFEUPDATE

**Input** Current sample $\boldsymbol{x}_t, y_t^f$, current best sample $\boldsymbol{x}^*, y^*$, current counters $c_s, c_t$ current trust region length $l_t$, success tolerance $\tau_s$, failure tolerance $\tau_f$, initial trust region length $l_0$, trust region limits $l_{\max}, l_{\min}$

1: **if** $y_t^g(\boldsymbol{x}_t) > 0$ and $y_t^f > y^*$ **then**
2: $\quad \boldsymbol{x}^*, y^* \leftarrow \boldsymbol{x}_t, y_t^f; c_s \leftarrow c_s + 1; c_f \leftarrow 0$
3: **else**
4: $\quad c_s \leftarrow 0; c_f \leftarrow c_f + 1$
5: **end if**
6: **if** $c_s = \tau_s$ **then**
7: $\quad l_t \leftarrow \min(2l_{t-1}, l_{\max}); c_s, c_f \leftarrow 0$
8: **else if** $c_f = \tau_f$ **then**
9: $\quad l_t \leftarrow \max(\frac{1}{2}l_{t-1}, l_{\min}); c_s, c_f \leftarrow 0$
10: $\quad$ **if** $l_t = l_{\min}$ **then**
11: $\quad\quad l_t \leftarrow l_0$
12: $\quad$ **end if**
13: **else**
14: $\quad l_t \leftarrow l_{t-1}$
15: **end if**

# 4 High-dimensional Safe Bayesian Optimization

In high-dimensional space, existing safe optimization methods are too conservative to efficiently optimize the objective function, and would be infeasible due to sparse discretization. Therefore we aim to improve sample efficiency by slightly relaxing the safety constraint to a probabilistic version. The probabilistic safety means each sample point is safe with a probability above predefined threshold $\alpha$, that is $\Pr(g(\boldsymbol{x}_t) \geq 0) \geq \alpha, \forall t \leq T$. In this sense, we introduce HDSAFEBO, an innovative algorithm designed for ensuring probabilistic safety while optimizing in a high-dimensional space. The workflow of this algorithm is illustrate in Figure 1 and Algorithm 1.

HDSAFEBO first uses the isometric encoder $\Pi$ to reduce the problem dimension, converting the original dataset from $\mathcal{X}$ to the low-dimensional latent space $\mathcal{Z} \in \mathbb{R}^d$ (Line 4). Isometry means the embedded subspace is able to preserve the distance of the original space according to the corresponding metric $d_{\mathcal{X}}, d_{\mathcal{Z}}$, i.e. $d_{\mathcal{X}}(\boldsymbol{x}, \boldsymbol{x}') = d_{\mathcal{Z}}(\Pi(\boldsymbol{x}), \Pi(\boldsymbol{x}'))$. Leveraging the historical record of function observations, we compute the posterior and confidence interval of both the objective and safety functions through distinct Gaussian processes. (Line 5). Then we define a local region to search over (Line 6), and identify the safe space within the local region using GP upper confidence bound (Line 7). We optimize the acquisition function over the safe space, and project the recommended latent point back to original input space using the decoder $\Pi^{-1}$ (line 8-9). Thompson sampling (TS) [52] was selected as the acquisition function $A$ due to its compatibility with the discrete nature of our safety estimation and search space, and its innate ability for batch optimization by sampling the GP posterior—an appropriate choice for high-dimensional tasks that support parallel evaluations. Finally the history data is updated via evaluating new inputs. (Line 10-12), and the local region parameters are updated based on the sample results (Line 13).

In conjunction with isometric embedding, we highlight two important components to improve the efficiency and safety: Optimistic Safety Identification and Local Search via Trust Region. For clear illustration, we first introduce algorithm details under identity mapping $\mathcal{I}(\boldsymbol{x}) = \boldsymbol{x}$, which is a special case of isometric embedding. In this way, the embedded subspace is equivalent to the original space $\mathcal{X}$, and inherits all the assumptions defined in Section 3.

## 4.1 Optimistic safety identification

To mitigate inefficiency concerns, HDSAFEBO distinguishes the safe region through the Gaussian process upper confidence bound of the safety function. This optimistic strategy allows for optimization over distinct safe regions and is viable in high-dimensional problems. In many real-world applications, the search space is inherently constrained by domain priors, where a certain proportion of decisions are safe even under random search. Therefore, we contend that optimistic safety identification is a more practical strategy. Here we derive the appropriate choice of the scalar $\beta_t$ to ensure step-wise probabilistic safety.

**Proposition 4.1.** *Let Assumptions 3.1 holds for the latent safety function $g$, and set $\beta_t$ satisfying $\Phi(\beta_t) \leq 1 - \alpha$. Then at every time step $t$:*

$$Pr(g(\boldsymbol{x}) \geq \mu_{t-1}(\boldsymbol{x}) + \beta_t \sigma_{t-1}(\boldsymbol{x})) \geq \alpha, \forall \boldsymbol{x} \in \mathcal{X}, \tag{3}$$

*where $\Phi(\cdot)$ is the cumulative distribution function (CDF) of the standard normal distribution $\mathcal{N}(0,1)$.*

Our use of $\beta_t$ here is different from the common-used setting [53], which would make $u_t(\boldsymbol{x}_t) > g(\boldsymbol{x})$ with a high probability and violate the probabilistic requirement. Our results indicate that the upper confidence bound is permissible when the probability threshold is less than $0.5$, representing the maximum acceptable level of safety violation. Otherwise, the algorithm reverts to a more conservative strategy, utilizing the lower confidence bound to identify safety. We also derive the upper bound of cumulative safety violation $V_T = \sum_{t=1}^{T} \max(0, -g(\boldsymbol{x}_t))$ of our optimistic strategy, where the proper choice of $\beta_t$ contributes to the reduction of safety violations:

**Theorem 4.2.** *Let assumptions 3.1 holds for safety function $g$. Define $\beta_T^c := 2log(|\mathcal{X}|T^2\pi^2/6\delta)$ and $C_1 := 8/\log(1 + \sigma^{-2})$. With probability at least $1 - \delta$, the sample points of Algorithm 1 at time step $T$ satisfy*

$$\mathbb{E}[V_T] \leq (1 - \alpha)\sqrt{C_1 T \beta_T^c \gamma_T}, \tag{4}$$

*where $\gamma_T$ is the maximum information gain for $g$ over $\mathcal{X}$.*

## 4.2 Local search via trust region

In addition to optimistic safety identification, a trust region method is employed to dynamically pinpoint local search regions, which has demonstrated impressive empirical performance over high-dimensional problems [43, 48, 7]. At each optimization round, a local search space is defined as a hypercube trust region around the current best safe point in the sample dataset. We design a safety sensitive strategy to update the trust region state, as illustrated in Algorithm 2. Specifically, a sampling round is considered "successful" if it finds a better reward while maintaining comprehensive safety (line 1-2). Conversely, it is labeled a "failure" if any unsafe points are found or if there is no discernible improvement (line 3-5). The side length is adjusted—increased for successes and decreased for failures—upon reaching a preset threshold (line 6-9). Unlike conventional local BO methods that discard all data and restart when the side length reaches its minimum, our approach resets $l_t$ to its initial length and retains all previous samples, ensuring a different, safer trajectory sampling than the initial instance (line 10-12).

We also provide the theoretical implications of our local search strategy in reducing the safety violations. The maximum information gain $\gamma_T$ is positively correlated with the size of search space $\mathcal{X}$. During the optimization, the actual search region of HDSAFEBO is restricted by the adaptive trust region, resulting in a lower maximum information gain compared to entire input space. Therefore the safety violation bound in Theorem 4.2 can be further reduced compared to global search.

## 4.3 Safe optimization with isometric embedding

In addition to identity mapping, when applying safe optimization over the subspace of other embeddings, a pertinent question arises regarding whether safety guarantees persist in the original space.

Here we demonstrate that, through the use of isometric embedding, the probabilistic safety guarantee in the embeded subspace can still be fulfilled in the original space when using stationary kernels.

**Proposition 4.3.** *Define* $d_\mathcal{X}(\boldsymbol{x}, \boldsymbol{x}') = |\boldsymbol{x} - \boldsymbol{x}'|, d_\mathcal{Z}(\boldsymbol{z}, \boldsymbol{z}') = |\boldsymbol{z} - \boldsymbol{z}'|$, *and* $u_t^\mathcal{Z}(\boldsymbol{z}) := \mu_{t-1}(\boldsymbol{z}) + \beta_t \sigma_{t-1}(\boldsymbol{z})$ *is the upper confidence bound estimated from GP over the embeded subspace* $\mathcal{Z}$. *Suppose the GP kernel is stationary and* $d_\mathcal{Z}(\boldsymbol{z}, \boldsymbol{z}') = d_\mathcal{X}(\Pi^{-1}(\boldsymbol{z}), \Pi^{-1}(\boldsymbol{z}'))$. *At every time step* $t$, *if the point* $\boldsymbol{z}$ *satisfies that* $Pr(u_t^\mathcal{Z}(\boldsymbol{z}) \geq 0) \geq \alpha$, *then we have* $Pr(g(\Pi^{-1}(\boldsymbol{z})) \geq 0) \geq \alpha$

The choice of the isometry embedding can be some linear mapppings, such as principal component analysis (PCA) when the number of principal components exceeds the effective dimension of the problem. Recent research efforts have also explored the utilization of deep neural networks to learn a subspace with both good representation and approximated distance-preserving property. We will show in experiment that HDSAFEBO is still able to maintain high safe probability when using approximated isometric embeddings.

# 5 Experiments

In this section, we first utilize synthetic functions to evaluate the algorithm performance. Then we apply HDSAFEBO to safely optimize the control of high-dimensional musculoskeletal system. Finally we demonstrate the potential of HDSAFEBO in optimizing neural stimulation induced human motion control through both simulation and real experiments. Additional experiments are presented in Appendix C.

**Evaluation metrics.** We assess the performance of the algorithm according to three metrics: the best feasible objective function value (Objective), the safe decision ratio of all samples (Safety), and the cumulative safety violation (Violation). The presented plots and tables display the means along with one standard error.

**Isometric embedding.** We utilize PCA in synthetic function and musculoskeletal system control tasks. In the neural stimulation induced human motion control task, we explore the use of isometric regularized variational autoencoder (IRVAE [54]) as the dimension reduction component.

## 5.1 Synthetic Function Optimization

To evaluate the algorithm performance under full assumption satisfaction, we sample both objective and safety functions from Gaussian processes, with an effective dimension $d_e = 40$, which is much lower than the input dimension $D = 1000$. We contrast the optimization performance with the following competitive algorithms target for high-dimensional constrained optimization: LineBO [24], SCBO [7], CONFIG [15], cEI [21] and CMA-ES [49]. We attempted to run SafeOpt [4], but it failed to expand the safe region from the initial points. We set PCA subspace dimension as $d = 50$. We show the optimization result in Table 1. HDSAFEBO achieves significantly better optimization performance, higher safety decision ratio, and lower cumulative safety violation compared against all baselines.

Table 1: Algorithm performance on GP synthetic functions. The total sample budget is 500 including 200 initial points. We show the averaged performance over 100 independent runs.

| Metric | HDSAFEBO | LineBO | SCBO | CONFIG | cEI | CMAES |
|---|---|---|---|---|---|---|
| Objective ($\uparrow$) | **3.96 $\pm$ 0.15** | 3.07 $\pm$ 0.04 | 2.97 $\pm$ 0.06 | 2.91 $\pm$ 0.05 | 2.9 $\pm$ 0.03 | 2.7 $\pm$ 0.04 |
| Safety ($\uparrow$) | **0.81 $\pm$ 0.02** | 0.78 $\pm$ 0.0 | 0.77 $\pm$ 0.0 | 0.77 $\pm$ 0.0 | 0.77 $\pm$ 0.0 | 0.78 $\pm$ 0.01 |
| Violation ($\downarrow$) | **27.42 $\pm$ 4.01** | 36.59 $\pm$ 0.82 | 39.05 $\pm$ 0.92 | 38.96 $\pm$ 0.96 | 39.61 $\pm$ 0.58 | 38.65 $\pm$ 1.89 |

## 5.2 Optimization for the Control of a Musculoskeletal System

We establish an upper limb control task utilizing a musculoskeletal system [55]. The objective is to optimize the activities of 55 hand-related muscles to rotate and hold a bottle in vertical position

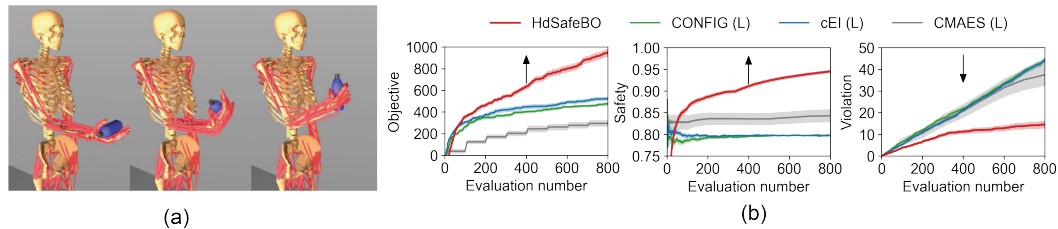

Figure 2: **Optimization for the control of a musculoskeletal system.** (a) Task illustration. (b) Optimization performance averaged over 50 independent runs. Arrows indicate the better direction.

(Figure 2 (a)). As in Mania et al. [56], we formulate the original reinforcement learning (RL) problem as a sampling problem, where the algorithms need to optimize a linear policy: $x \in \mathbb{R}^{|a| \times |o|}$, where $|a| = 55$ and $|o| = 65$ are the dimensions of the action space (number of muscles) and the observation space, respectively. The policy to be optimized has $D = 3575$ parameters, making it a very high-dimensional task. The objective function is set as the accumulated reward from the environment, and the safety function is defined as the landing speed of the bottle (see Appendix B.5). We collected muscle activation and use PCA to build the muscle synergies of performing the task, reducing action dimension from 55 to 5 (3575 to 325 for policy dimension). While the search space is significantly reduced, the remaining optimization problem is still high-dimensional.

We find that *all baselines fail to improve the objective when optimizing over the original parameter space*. Therefore we conducted baseline runs over the subspace derived from PCA (denoted with (L)), where dimension reduction facilitated effective optimization. The optimization results are presented in Figure 2 (b). SCBO (L) and LineBO (L) are omitted from the figure as they fail to attain a positive reward. We observe even within the PCA subspace, all the shown baselines demonstrated lower efficiency and sampled more unsafe parameters compared to HDSAFEBO. Our proposed method stands as the pioneering algorithm to achieve efficient and safe optimization over high-dimensional musculoskeletal system control. We also conducted an ablation study on the components of HDSAFEBO in Appendix C.2, where both optimistic safe identification and local search were shown to contribute to a safer optimization process.

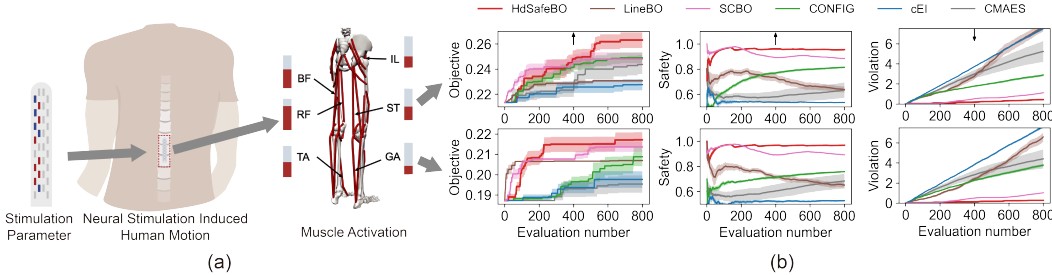

Figure 3: **Optimization for the control of neural stimulation induced human motion**. (a) Task illustration. IL, RF, TA, BF, ST, GA are different group of target muscle on the lower limb. (b) Optimization performance on the control of neuromuscular model for semitendinosus (ST), and gastrocnemius (GA), averaged over 10 independent runs. Arrows indicate the better direction.

### 5.3 Optimization for the Control of Neural Stimulation Induced Human Motion

Apart from direct activation of muscle-tendon units, locomotion of musculoskeletal systems can also be governed by muscle synergies arising from spinal cord stimulation, as observed in the central nervous system of vertebrates and certain neuromuscular robotic designs [57, 58]. While the neuromuscular system holds the potential to achieve robust control performance with higher energy efficiency compared to direct muscle actuation, the mapping from stimulation inputs to motion becomes less straightforward under complex neural systems. In this section, we showcase the success of HDSAFEBO in optimizing the control of intricate neural stimulation-actuated human motion through simulation and real experiments.

As shown in Figure 3 (a), we want to improve the control of lower limb muscles via an electrode array implanted in the human spinal cord. By setting different parameters, the electrical stimulation delivered by the electrode array can induce patient's muscle activities, allowing us to control the lower limb movements of the patient. The stimulation parameter space consists of 16-contact spatial configuration (discrete) and current intensity (continuous). Our optimization goal is the selectivity index of target muscle groups (see Appendix B.6). A higher selectivity index indicates better control over the target muscle group and less influence over non-target muscles. To enhance the physical interpretability of discrete input representation, we transformed 17-dimensional vectors into an electric field map with $52 \times 14$ pixels using simplified computation. Subsequently, we generate unlabeled synthetic data with clinical priors and train an IRVAE with a 16-dimensional latent space to embed and reconstruct the electric field map, with distance-preserving checks in Appendix C.3.

**Simulation over neuromuscular model**. In simulation, we use a human neuromuscular model as the function oracle, which is capable of computing the evoked electric field around the spinal cord given certain stimulation parameter, and inferring the lower limb muscle activation (see Appendix B.6). The maximum induced muscle activation is used as the safety function to avoid causing harm to the patient during optimization. We choose to optimize the selectivity of emitendinosus (ST) and gastrocnemius (GA) for their importance during human walking. The simulation results are depicted in Figure 3 (b). Despite the subspace space dimensions being comparable to the original space, HDSAFEBO achieves safe exploration by optimizing on the continuous manifold, which obtains the best control performance while maintaining the highest safety selection ratio and lowest cumulative safety violation compared to other algorithms.

**Real-world experiments on paraplegic patient**. We further applied HDSAFEBO to improve the motor function of a paraplegic patient with the same electrode array implanted. Starting with 132 initial configurations from clinical prior, a total of 504 additional trials were conducted with the patient over a 1-month period. We observed selectivity improvement of 7 out of 8 target muscles compared to the baseline (left IL: 0.112, left RF: 0.143, left TA: 0.097, left BF: 0.380, right IL: 0.00, right RF: 0.266, right TA: 0.216, right BF: 0.141). During the whole experimental procedure, only three configurations recommended by HDSAFEBO were rated as unsafe, which evoked large lower limb movements but no physical damage or pain. Our real-world experiment underscores the practicality of HDSAFEBO in safely optimize the control of complex neuromuscular systems.

## 6 Conclusion

We develop HDSAFEBO for optimizing of the control over high-dimensional embodied systems under safety constraints. Our proposed method employs a local optimistic safe strategy to optimize the objective function and expand the safe region, with probabilistic safety guarantee and cumulative safety violation bound. HDSAFEBO is able to optimize high-dimensional input ranging from a few dozen to several thousand variables with safety guarantee. The algorithm can efficiently optimize the control of high-dimensional human musculoskeletal systems with high safety probability, and successfully optimize human motion control via neural stimulation in real clinical experiments. HDSAFEBO has great potential to safely optimize the control of real-world high-dimensional embodied systems online.

**Limitations**. While we provide the probabilistic safety guarantee for HDSAFEBO, real-world applications may fail to fully satisfy the theoretical assumptions regarding function regularity. The imperfection of the trained embedding could lead to unsafe behavior when optimizing over a reduced subspace. It is important to pre-check the distance-preserving quality of this subspace before conducting online optimization, and improvements can be made by synthesizing additional unsupervised data based on domain knowledge. The parameter space in our real human experiment has been restricted by domain prior. Directly applying HDSAFEBO to a completely unexplored problem may cause more unsafe decisions due to its optimism to safety. The total evaluation number of HDSAFEBO is constrained by cubic complexity of Gaussian process. The computational complexity prevents the use of HDSAFEBO in very long time horizon optimization problems.

**Acknowledgments**

This project is funded by STI 2030-Major Projects 2022ZD0209400 and Tsinghua DuShi fund.

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

# A Theoretical Analysis

## A.1 Proof of Proposition 4.1

*Proof.* Fix $t \geq 1$ and $\boldsymbol{z} \in \mathcal{Z}$. Conditioned on $\boldsymbol{y}_{t-1} = (y_1^g, \ldots, y_{t-1}^g), \{\boldsymbol{z}_1, \ldots, \boldsymbol{z}_{t-1}\}$ are deterministic, and $g(\boldsymbol{z}) \sim \mathcal{N}(\mu_{t-1}(\boldsymbol{z}), \sigma_{t-1}(\boldsymbol{z})))$. For $r \sim \mathcal{N}(0,1)$, $\Pr(r > c) = 1 - \Phi(c)$.

Therefore, by applying $r = (g(\boldsymbol{z}) - \mu_{t-1}(\boldsymbol{z}))/\sigma_{t-1}(\boldsymbol{z})$ and $c = \beta_t$, the statement holds. □

## A.2 Proof of Theorem 4.2

We first introduce the results in Chowdhury and Gopalan [59] which prove the selection of confidence scalar to contain the function with high probability, and the upper bound of cumulative standard deviations.

**Lemma A.1.** *(Lemma 5.1 in Srinivas et al. [53]) Let assumptions 3.1 and ?? hold for g. For any $\delta \in (0,1)$, with probability at least $1 - \delta$, the following holds for all $x \in \mathcal{X}$ and $1 \leq t \leq T, T \in \mathbb{N}$,*

$$|g(\boldsymbol{x}) - \mu_{t-1}(\boldsymbol{x})| \leq (\beta_t^c)^{\frac{1}{2}} \sigma_{t-1}(\boldsymbol{x}), \tag{5}$$

*where $\beta_t^c = 2 log(|\mathcal{X}|t^2\pi^2/6\delta)$*

Then we can bound the instantaneous safety violation of HDSAFEBO.

**Lemma A.2.** *Let assumptions 3.1 hold for safety function g. With probability at least $1 - \delta$, the sample points of Algorithm 1 for all time steps $1 \leq t \leq T$ satisfy*

$$\mathbb{E}[v_t] = \mathbb{E}[\max(0, -g(\boldsymbol{x}_t))] \leq 2(1-\alpha)(\beta_t^c)^{\frac{1}{2}}\sigma_{t-1}(\boldsymbol{x}_t). \tag{6}$$

*Proof.* We denote $[\cdot]^+ := \max(\cdot, 0)$. Then we have

$$\mathbb{E}[v_t] = \mathbb{E}[[-g(\boldsymbol{x}_t)]^+] \tag{7}$$
$$= \mathbb{E}[[-g(\boldsymbol{x}_t) - u_t(\boldsymbol{x}_t) + u_t(\boldsymbol{x}_t)]^+] \tag{8}$$
$$\leq \mathbb{E}[[-g(\boldsymbol{x}_t) + u_t(\boldsymbol{x}_t)]^+ + [-u_t(\boldsymbol{x}_t)]^+] \tag{9}$$
$$= \mathbb{E}[[-g(\boldsymbol{x}_t) + u_t(\boldsymbol{x}_t)]^+] \tag{10}$$
$$= \mathbb{E}[[-g(\boldsymbol{x}_t) + u_t(\boldsymbol{x}_t)]^+ \mathbb{1}\{g(\boldsymbol{x}_t) \geq u_t(\boldsymbol{x}_t)\} + [-g(\boldsymbol{x}_t) + u_t(\boldsymbol{x}_t)]^+ \mathbb{1}\{g(\boldsymbol{x}_t) < u_t(\boldsymbol{x}_t)\}] \tag{11}$$
$$= (1-\alpha)[-g(\boldsymbol{x}_t) + u_t(\boldsymbol{x}_t)]^+ \tag{12}$$
$$\leq (1-\alpha)[-(\mu_{t-1}(\boldsymbol{x}_t) - (\beta_t^c)^{\frac{1}{2}}\sigma_{t-1}(\boldsymbol{x}_t)) + \mu_{t-1}(\boldsymbol{x}_t) + (\beta_t^c)^{\frac{1}{2}}\sigma_{t-1}(\boldsymbol{x}_t)] \tag{13}$$
$$= 2(1-\alpha)(\beta_t^c)^{\frac{1}{2}}\sigma_{t-1}(\boldsymbol{x}_t), \tag{14}$$

where the inequality (9) follows by the fact that $[a + b]^+ \leq [a]^+ + [b]^+, \forall a, b \in \mathbb{R}$, the equality (10) is derived from the fact that $u_t(\boldsymbol{x}_t) \geq 0$, the equality (12) is derived from Proposition 4.1, and the inequality (13) is derived from Lemma A.1.

□

**Lemma A.3.** *(Theorem 1 in Srinivas et al. [53].) Let $\boldsymbol{x}_1, \cdots, \boldsymbol{x}_T$ be the points selected by the algorithms. With $C_1 := 8/\log(1 + \sigma^{-2})$,*

$$\sum_{t=1}^{T} 2(\beta_t^c)^{\frac{1}{2}}\sigma_{t-1}(\boldsymbol{x}_t) \leq \sqrt{C_1 T \beta_T^c \gamma_T} \tag{15}$$

Finally we bound the summation of instantaneous safety violations.

*Proof.*

$$\mathbb{E}[V_T] = \mathbb{E}[\sum_{t=1}^{T} v_t] \tag{16}$$

$$= \sum_{t=1}^{T} \mathbb{E}[v_t] \tag{17}$$

$$\leq \sum_{t=1}^{T} 2(1-\alpha)(\beta_t^c)^{\frac{1}{2}} \sigma_{t-1}(\boldsymbol{x}_t) \tag{18}$$

$$\leq \sum_{t=1}^{T} 2(1-\alpha)(\beta_T^c)^{\frac{1}{2}} \sigma_{t-1}(\boldsymbol{x}_t) \tag{19}$$

$$\leq (1-\alpha)\sqrt{C_1 T \beta_T^c \gamma_T}, \tag{20}$$

where the inequality (18) follows by Lemma A.2, the inequality (19) follows by the monotonicity of $\beta_t^c$, and the inequality (20) follows by Lemma A.3. $\qquad\square$

### A.3 Proof of Proposition 4.3

*Proof.* Define $k^{\mathcal{X}}, k^{\mathcal{Z}}$ are the kernel function of GP with same hyperparameter over $\mathcal{X}$ and $\mathcal{Z}$ respectively. For stationary kernel, the kernel function value between $\boldsymbol{x}$ and $\boldsymbol{x}'$ is only depend on $|\boldsymbol{x} - \boldsymbol{x}'|$, therefore we can write the $k^{\mathcal{X}}, k^{\mathcal{Z}}$ as a function of metric $d_{\mathcal{X}}, d_{\mathcal{Z}}$:

$$k^{\mathcal{X}}(\boldsymbol{x}, \boldsymbol{x}') = k^{\mathcal{X}}(d_{\mathcal{X}}(\boldsymbol{x}, \boldsymbol{x}')), k^{\mathcal{Z}}(\boldsymbol{z}, \boldsymbol{z}') = k^{\mathcal{Z}}(d_{\mathcal{Z}}(\boldsymbol{z}, \boldsymbol{z}')). \tag{21}$$

We define $k^{\mathcal{X}}$ as the kernel function of GP over $\mathcal{X}$, and $u_t^{\mathcal{X}}$ as the estimated upper confidence bound of GP over $\mathcal{X}$. Utilizing the property of isometric embedding, we can obtain equivalent GP estimation between original space and embedded subspace:

$$k^{\mathcal{Z}}(d_{\mathcal{Z}}(\boldsymbol{z}, \boldsymbol{z}')) = k^{\mathcal{X}}(d_{\mathcal{X}}(\Pi^{-1}(\boldsymbol{z}), \Pi^{-1}(\boldsymbol{z}'))). \tag{22}$$

Therefore the estimated upper confidence bound is also equivalent. For every sample $\boldsymbol{z}$ from HDSAFEBO, we have

$$\Pr(g(\Pi^{-1}(\boldsymbol{z})) \geq 0) \geq \Pr(u_t^{\mathcal{X}}(\Pi^{-1}(\boldsymbol{z})) \geq 0) \tag{23}$$

$$= \Pr(u_t^{\mathcal{Z}}(\boldsymbol{z}) \geq 0) \tag{24}$$

$$\geq \alpha, \tag{25}$$

the statement holds.

$\qquad\square$

## B    Experimental Details

The full implementation of our experiments can be found on our project page: https://lnsgroup.cc/research/hdsafebo. Our musculoskeletal model will be released soon. In the meantime, the model can be accessed for research purposes upon request (ysui@tsinghua.edu.cn).

### B.1    PCA training

We employ PCA using the scikit-learn library [1] with default parameter settings.

---
[1] https://scikit-learn.org/stable/modules/generated/sklearn.decomposition.PCA.html

## B.2 Autoencoder training

We employ IRVAE on neural stimulation task and digital generation task by directly using the paper's original repository[2]. We use MLP as the VAE module for all tasks, and list the model detail in Table 2. We train all models for 300 epochs using Adam[60] optimizer with a learning rate of 0.0001.

| Task | Layer number | Hiddien number | Latent dimension |
|------|------|------|------|
| Neural stimulation | 4 | 256 | 16 |
| Digital generation | 4 | 256 | 6 |

Table 2: Autoencoder model detail.

## B.3 Algorithm Implementation

For implementation of HDSAFEBO, We use BoTorch as the GP inference component [61]. We also use BoTorch to replicate LineBO, SCBO, CONFIG, cEI and cEI-Prob. For LineBO, we choose to implement the random line embedding version shown in the main paper. We use the package pycma[3] to run CMA-ES on benchmarks.

All GP-based methods uses matérn kernel and fits kernel parameters after each iteration. During the experiment, we set $\tau_s, \tau_f, l_0, l_{\max}, l_{\min}$ for HDSAFEBO and SCBO (the default setting of SCBO). We use Thompson sampling for LineBO and CONFIG, and use inherent acquisition function for other BO baselines. Confidence scalar $\beta$ is set as 2 for HDSAFEBO, LineBO and CONFIG across all experiments. We set the latent optimization bound as the upper bound and lower bound of training points in the latent space. Other baseline parameters are set to default values as in the original implementation.

During the experiment, we set the sample size to 10 for all tasks in the main paper, and sample one point each iteration in constrained digital generation task.

## B.4 Synthetic Function

We sample objective and safety function from separate Gaussian process with Matérn kernel and length scale as 0.05, which is implemented using GPyTorch[4]. We set the safety threshold to $-0.75$. For each independent run, we use a random linear projection $\Pi_{\mathrm{rand}} \in \mathbb{R}^{D \times d}$ to create $d$-dimensional latent space, and randomly select $d_e$ variables as the function effective dimension.

## B.5 Musculoskeletal Model Control

We use a full-body musculoskeletal model which actuates locomotion by controlling muscle activation. Here we only control the right hand part (below elbow), and fix other joints, leading to 55 muscles and 28 joints. The primary task is to control hand muscles to maintain a steady vertical grip on a bottle. At the beginning of the episode, the bottle is initially positioned horizontally in the hand. The initial task involves first rotating the bottle to achieve and maintain a vertical orientation. At each time step, the reward from the environment is computed as follows:

$$r = r_{\mathrm{pose}} + r_{\mathrm{bonus}} - 10 * r_{\mathrm{penalty}} + r_{\mathrm{grasp}} + 2 * r_{\mathrm{survive}} - r_{\mathrm{activation}} - 100 * r_{\mathrm{drop}} \quad (26)$$

where $r_{\mathrm{pose}}$ is the difference between the bottle and vertical orientation, computed by Euler angle. $r_{\mathrm{bonus}}$ is the reward when the difference falls below a predefined threshold. $r_{\mathrm{penalty}}$ is positive when the bottle position is out of from the predefined range. $r_{\mathrm{grasp}}$ is the distance between the centroid of

---

[2]https://github.com/Gabe-YHLee/IRVAE-public
[3]https://github.com/CMA-ES/pycma
[4]https://gpytorch.ai/

the bottle and palm joints. $r_{\text{survive}}$ is the reward for not dropping the bottle during the current time step. $r_{\text{activation}}$ is the penalty for large muscle activations. $r_{\text{drop}}$ is the penalty when the bottle drop from hand. The overall simulation is based on Mujoco [62].

When the height of the bottle is below 0.4m, we consider it to be dropped from the hand and the episode concludes. We record the speed of episode ending as the landing speed of the bottle. We use a safety threshold of 4.4, which is the average landing speed when randomly sampling the environment. We train a Soft Actor-Critic agent [63] for 150k time steps under different observation setting, and rollout for 1000 episode to build a muscle activation dataset with 107, 439 datapoints.

The performance video of HDSAFEBO and other baselines (in the subspace from PCA) is shown in the supplementary file. HDSAFEBO is capable of quickly rotating the bottle and hold vertically and steadily, while other algorithms either hold the bottle non-vertically, or learn drop the bottle safely to avoid penalty of wrong orientation.

### B.6  Modeling of the human neuromuscular system

We developed an average model of the human spinal cord based on anatomical statistics(model paper under review, [64, 65, 66, 67, 68]). The model contains gray matter, white matter, nerve roots, cerebrospinal fluid (CSF), and dura mater of T12-S2 segments of the spinal cord which are related with the motor control of lower limbs. The specific conductivity values of the modeled tissues were set refer to [69]. Electric fields induced by different stimulation parameters were derived using finite element method (FEM). To calculate the stimulation effects for different muscles, we redistricted the cord model according to reported results of the segmental innervation for lower limb muscles([70, 71]). Six groups of muscles of bilateral lower limbs were studied: iliopsoas (IL), vastus lateralis and rectus femoris (VL&RF), tibialis anterior (TA), biceps femoris muscle and gluteus maximus (BF&GM), semitendinosus (ST), and gastrocnemius (GA).

To evaluate the selectivity of stimulation for certain muscle, we used a selectivity index (SI) to characterize the distribution of the electric field. The selectivity index for the $i$th muscle was defined as follows:

$$\text{SI}_\text{i} = \mu_i - \frac{1}{m_{\text{neighbor}} - 1} \sum_{j \neq i}^{m_{\text{neighbor}}} \mu_j \tag{27}$$

where $m_{\text{neighbor}}$ represents the number of muscles whose motor neuron pools are adjacent to the $i$th muscle's. The selectivity index ranges from -1 to 1, where -1 represents the maximum of activation of all undesired muscles with a complete absence of activation of the targeted muscle, 0 indicates that all muscles are activated at the same level, and 1 means the targeted muscle is activated at the greatest extent while no undesired muscles are activated. And $\mu_i$ is the normalized activation of the $i$th muscle and is defined as follows in the simulation:

$$\mu_i = \frac{\iiint_{\Omega_i} f(x,y,z)\,\mathrm{d}x\mathrm{d}y\mathrm{d}z}{\iiint_{\Omega_i} 1\,\mathrm{d}x\mathrm{d}y\mathrm{d}z} \tag{28}$$

$$f(x,y,z) = \begin{cases} 1, & \text{if } AF(x,y,z) > AF_{\text{threshold}} \\ 0, & \text{if } AF(x,y,z) \leq AF_{\text{threshold}} \end{cases} \tag{29}$$

$\Omega_i$ is the segmental volume of the $i$th muscle in the cord. $AF$ is the activating function, defined as the second spatial derivative of extracellular voltage along an axon([72, 73]).

We use the spinal model to traverse all stimulation parameters with 1 cathode with anodes no more than 3, and 2 cathodes with anodes no more than 2, leading to a spinal cord stimulation (SCS) dataset with 218,000 stimulation parameters and predicted muscle activation. We compute the objective function using 27, and compute the safety function as $g(x) = 1 - max_i(\mu_i)$. The selectivity index distribution is shown in Figure 4. We set the safe threshold as 0.05, with nearly half of the traversed parameters are safe.

We convert electrode parameters from 17d vector to a 2d electric field image using simplified computation. In concrete, we map contact combinations to the spatial position in the electrode, linearly compute the diffusion of electrical field from each cathode and anode, and multiply the map by current intensity. One example of generated electric field image is shown in Figure 5.

We use the constructed SCS dataset as the function oracle. For a given 2d map, we set its function value as the function value of the nearest unevaluated point in the dataset measured by the electrical field map.

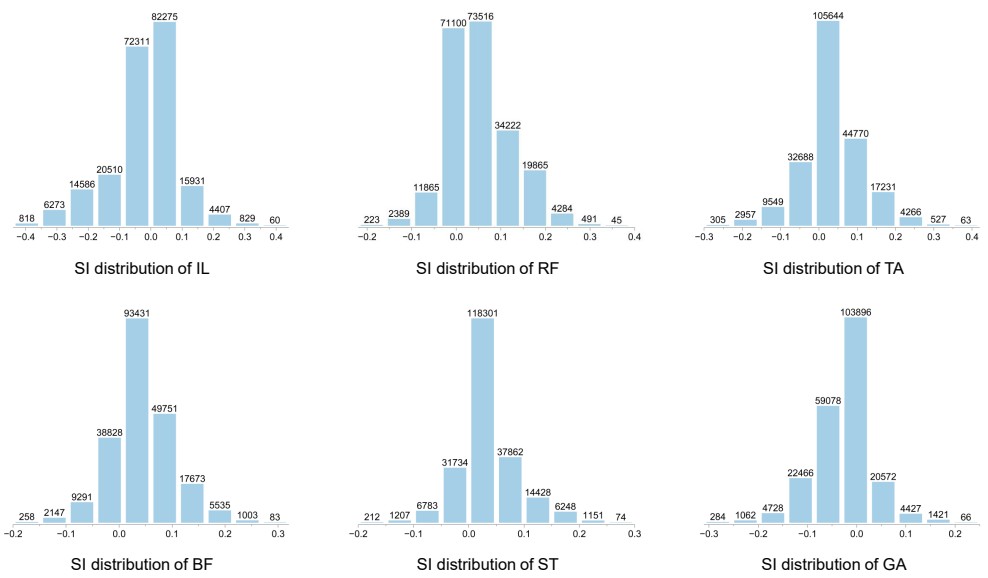

Figure 4: Distribution of SI for six muscle groups of different configurations used in SCS simulation experiment

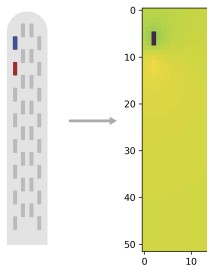

Figure 5: 2d electrical map computation.

## B.7 Clinical experiment of neuromuscular system control

We employ HDSAFEBO in the treatment of spinal cord stimulation to find more selective stimulation parameters for different muscles. The clinical experiments received approval from the Institutional Review Board (IRB) of the hospital. All the trials were conducted under the supervision of therapists. The patient was seated in the wheelchair in a comfortable way and was told to relax. During the first period, typical parameters which were usually used in the therapy (e.g. bipolar stimulation) were delivered to the patient while the evoked muscle activities were recorded using EMG. These data (132 trials) were used to initialize HDSAFEBO. Except for the first 132 trials as the initial data, 441 out of 504 trials are recommended by HDSAFEBO. The other trials were conducted by the therapist.

We focus on 8 groups of muscles: iliopsoas (IL), rectus femoris (RF), tibialis anterior (TA), and biceps femoris (BF) for both sides. The clinical selectivity index was defined as following:

$$\text{SI}_i = \frac{\mu_i}{1 + \sum_{j \neq i}^{m} \mu_j} \tag{30}$$

where $\mu_i$ represents the normalized peak-to-peak value of the evoked EMG for the $i$th muscle and $m$ is the total number of the target muscles.

During the optimization, We restrict the number of cathodes and anodes to only evaluate configurations consistent with clinical priors. We used the Electromyography (EMG) to compute the selectivity index and queried safety scores from the patient and the therapists. For each trial, our algorithm recommended the parameter based on the history data and configured it onto the stimulator, which would deliver electrical stimulation to the patient for 800 ms at a frequency of 10 Hz. Peak-to-peak values were averaged and normalized to obtain selectivity indices of different muscles after stimulation. The calculated feedback and queried safety score were used to update the optimizer and it would recommend a new parameter. The tasks were refined sequentially and all the history data were reused when optimizing a new task.

## C    Additional Experiment

### C.1    Baselines performance over embedded subspace

We run baselines on the same embedded space as HdSafeBO in optimizing GP synthetic functions (Table 3). We observe that, overall, the optimization and safety performances slightly improved. However, HdSafeBO still outperforms these baselines, as the reduced space remains too high-dimensional for them.

Table 3: Algorithm performance on GP synthetic functions. We show the averaged performance of 500 evaluations over 100 independent runs. "L" indicates baselines optimize over the embedded latent space, "O" indicates baselines optimize over the original input space.

| Method | HdSafeBO | LineBO | | SCBO | | CONFIG | | cEI | | CMAES | |
|---|---|---|---|---|---|---|---|---|---|---|---|
| | | L | O | L | O | L | O | L | O | L | O |
| Objective (↑) | **3.96 ± 0.15** | 3.03 ± 0.05 | 3.07 ± 0.04 | 3.15 ± 0.06 | 2.95 ± 0.04 | 2.99 ± 0.04 | 2.91 ± 0.05 | 2.92 ± 0.03 | 2.9 ± 0.03 | 2.69 ± 0.04 | 2.7 ± 0.04 |
| Safety (↑) | **0.81 ± 0.02** | 0.79 ± 0.0 | 0.78 ± 0.0 | 0.77 ± 0.0 | 0.77 ± 0.0 | 0.77 ± 0.0 | 0.77 ± 0.0 | 0.77 ± 0.0 | 0.77 ± 0.0 | 0.77 ± 0.01 | 0.78 ± 0.01 |
| Violation (↓) | **27.42 ± 4.01** | 36.26 ± 1.28 | 36.59 ± 0.82 | 38.51 ± 0.64 | 38.47 ± 0.69 | 39.08 ± 0.6 | 38.96 ± 0.96 | 39.15 ± 0.64 | 39.61 ± 0.58 | 38.01 ± 1.94 | 38.65 ± 1.89 |

### C.2    Ablation on algorithm components

In the musculoskeletal system control task, we conducted an ablation study of two components in HdSafeBO: local search and optimistic safe identification (Figure 6). We observed that without local search, the algorithm tends to over-explore, leading to degraded optimization and safety performance in this high-dimensional problem. Without optimistic safe identification, the algorithm makes more unsafe selections during the early stages of optimization. Combining these two components enables a safe and efficient optimization procedure.

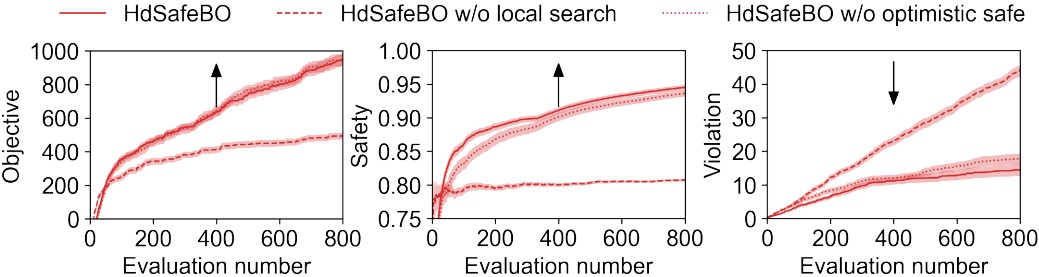

Figure 6: Ablation on components of HdSafeBO. Optimization performance averaged over 50 independent runs.

## C.3 Distance Preserving of IRVAE

In the implementation of HDSAFEBO, we explore the use of IRVAE to learn a mapping with distance-preserving property. We randomly sample 10,000 datapoints from SCS dataset. Using conventional VAE and IRVAE trained from same dataset, we compute the point-wise distance as well as GP estimation difference between the original and latent space, as shown in Table 4. The results

Using the 218,000 synthetic training data in human neural stimulation control, we plotted the pairwise distance of points in both original input space and latent space learns by IRVAE (Figure 7). We can observe that the learned embedding space exhibits approximately scaled isometry.

| Model | Linear correlation of distance (↑) | GP mean estimation difference (↓) | GP variance estimation difference (↓) |
|---|---|---|---|
| IRVAE | **0.9729** | **1.2936** | **2.9953** |
| conventional VAE | 0.8923 | 2.1839 | 4.5608 |

Table 4: Distance preserving comparison between IRVAE and conventional VAE

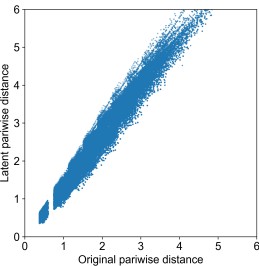

Figure 7: Pair-wise Distance of training data points in neural stimulation induced human motion control task.

## C.4 Ablation on Confidence Bound Scalar

Here we run HDSAFEBO with $\beta = 0, 2, 4, 8, 16$ on musculoskeletal model control task, and shown the results in Figure 8. The safety and violation metric may become slightly worse as $\beta$ increases. We observe the algorithm performance is similar under a wide range choice of $\beta$ on this high-dimensional task.

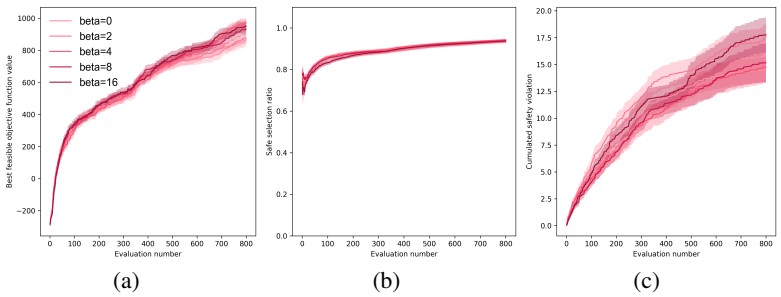

(a)                    (b)                    (c)

Figure 8: Ablation study on confidence bound scalar $\beta$.

## C.5 Comparison with random-embedding BO

We additionally run HesBO and BAxUS on the musculoskeletal model control task and the neural stimulation task with no safety constraints in simulation. Due to the algorithmic mechanism of HesBO and BAxUS, we cannot directly use the same initial point as HDSAFEBO. Therefore we randomly sample initial points from their corresponding latent space. In HesBO, we set the same latent dimension number as in HDSAFEBO. Table 5 shows the best objective function values found by algorithms (shown as mean $\pm$ 1 std).

Table 5: Best objective function values found by algorithms.

| Algorithm | SCS-ST | SCS-GA |
|-----------|--------|--------|
| HDSAFEBO | **0.26 ± 0.02** | **0.22 ± 0.01** |
| HesBO | 0.23 ± 0.0 | 0.19 ± 0.02 |
| BAxUS | 0.24 ± 0.03 | 0.19 ± 0.05 |

We observe HDSAFEBO still outperforms HesBO and BAxUS across all tasks, even when optimizing under safety constraint. We think using IRVAE enables utilizing the pre-collected unlabeled data to learn a better representation than random projection.

## C.6 Hand-writing Digital Generation

In hand-writing digital generation task, the goal is to generate images of target digit as thick as possible, while keeping the image valid for the required number. Using this task we can test the algorithm performances when the latent dimension is low. We trained a fully-connected IRVAE with a latent dimension of 6 and use a two-layer CNN model as the predictor. We set the objective function as the sum of image pixel intensities and the safety function as the prediction probability of target number. Since the CNN prediction is very sharp, we wrap the CNN output via a softmax layer with temperature as 200. We set the sample budget as 200 including with 20 images of target digit as the initial data.

We summarize the averaging performance in Table 6. We observed HDSAFEBO outperforms all baselines in terms of optimization performance and safety violation. Note that while HDSAFEBO efficiently finds highest objective, its safety violations is 63% less than the second best method original SCBO.

Table 6: Experiment results of constrained hand-writing digital generation. We evaluate algorithm performance of generating digital from 0 to 9 in terms of best found feasible objective value (higher is better) and cumulative safety violation (lower is better). Objective values are normalized by best feasible point in the MNIST dataset. The results are shown as mean performance ± one standard deviation across ten tasks.

| Metric | HDSAFEBO | SCBO | CONFIG | cEI | CMAES |
|--------|----------|------|--------|-----|-------|
| Objective | **1.14 ± 0.12** | 1.08 ± 0.18 | 0.77 ± 0.26 | 0.77 ± 0.26 | 0.68 ± 0.12 |
| Violation | **18.99 ± 10.82** | 52.01 ± 20.28 | 72.09 ± 17.85 | 72.04 ± 17.82 | 72.01 ± 7.06 |

## C.7 Run time performance

We ran HdSafeBO and baseline algorithms on the human neural stimulation control problem and recorded the average run-time of each iteration in Table 7. Most baseline algorithms have a processing time of around 10 seconds.

In our problem setting, we consider the run-time difference to be marginal compared to the actual experiment time for each trial. For instance, in our real experiments, applying recommended stimulation parameters typically takes 1-2 minutes. Therefore, we compared optimization performances based on the number of evaluations.

| Method | HdSafeBO | LineBO | SCBO | CONFIG | cEI | CMAES |
|--------|----------|--------|------|--------|-----|-------|
| Iteration time | 11.75 ± 1.35s | 9.88 ± 1.01s | 11.99 ± 1.21s | 9.49 ± 1.05s | 20.38 ± 1.21s | 9.59 ± 0.72s |

Table 7: Run-time per iteration in neural induced human motion control task. Results show mean ± 1 std averaged over 80 iterations.

# D Additional Related Work

Here we additionally discuss more works about high-dimensional Bayesian optimization besides dimension-reduction based BO and local BO.

Due to the inversion of kernel function matrix, the complexity of GP inference scales exponentially with the sample number, limiting the search budget of high-dimensional problems. Sparse GP or variational GP is used to achieve scalable sampling over the high-dimensional space [74, 75, 76].

Another line of work assume the addictive structure of the objective functions, and decomposes the function to solve the low-dimensional sub-problem decentrally[77, 78, 79, 80].

To overcome over-exploration issue over the high-dimensional space, several works also propose to shape kernel prior to sample points near the search center [42, 81].

