# OpenReview forum: "Safe Bayesian Optimization for the Control of High-Dimensional Embodied Systems"
_robot-learning.org/CoRL/2024/Conference — CoRL 2024_

### Official Review · Reviewer_kViv · 2024-07-15
**Safe Bayes Opt Corl 2024**

**Originality:** 3
**Technical Quality:** 3
**Clarity Of Presentation:** 2
**Potential Impact:** 3
**Recommendation:** 3
**Confidence:** 4

**Review:**

Strengths:
- The authors introduce a novel method that allows to optimize high-dimensional control inputs to robotics systems with high safety probability, specifically targeting musculoskeletal systems.
- The inclusion of both simulated and real-world examples, such as the control of an upper limb with 55 muscles and neural stimulation of a lower limb in a paraplegic patient, enhances the practical relevance and impact of the proposed method.


Weaknesses:


- Only one derivative-free algorithm that is not a BO method (CMA-ES) has been used for the comparison.
- Clarification Needed: In Figure 2b, there is an inconsistency regarding whether the comparison with HdSafeBO was conducted over the subspace (L) derived from PCA. This needs clarification.
- Safety Constraints Definition: The specific safety constraints should be more clearly defined and justified.
- The focus on Bayesian optimization methods could be better contextualized within the broader scope of optimization techniques.
Please also see questions for rebuttal below

**Quality Of The Limitations Section:**

2

**Questions For Rebuttal:**

-	In Fig. 2b, the performance of the optimization is shown. It is unclear whether the comparison with HdSafeBO was also conducted over the subspace (L) derived from PCA. The text (line 232) mentions this, but the figure does not have the subspace denotation L? Please clarify this inconsistency.
-	Additionally, if HdSafeBO was not run over the subspace derived from PCA, it would be valuable to see this comparison to understand he influence of dimension reduction.
-	What are the specific safety constraints in your study? Landing speed of the bottle/ speed of bottle at the end of the episode? In the 2nd case: Is the maximum induced muscle activation the constraint? If so, isn't this an input parameter to the model? Would it not be more meaningful to characterize maximum joint angle velocities or joint violations as measures for pain and wear/tear of the system?
-	Regarding the fairness of comparison, can you provide an estimation of the run-time comparison between the algorithms? In Fig. 2 and 3 you plot the (commonly used) performance over the evaluation number. However, in real robotics applications, the actual time to optimize is crucial. If I can run twice as many optimizations with LineBO compared to HdSafeBO in the same time, the difference observed in Figure 3 might be less pronounced. Could you address this aspect?
-	In Chapter 2.2, the title claims "high-dimensional BO," but the last paragraph also discusses CMA-ES as another method. This seems inconsistent. Additionally, the focus of the comparison is heavily on Bayesian optimization methods, while the introduction generally states that BO has advantages for optimizing black-box functions, among which many other methods exist. Could you clarify the reasoning behind this focus?
-	How does your method differ / outperform GOSAFEOPT that has been also used to safely tune legged locomotion controllers for high-dimensional dynamical systems using safe BO?
-	Concerning code and model availability, Reference 55 refers to a model that is not openly available. Please ensure that both the model and the code are provided as open-source upon acceptance.

**Small comments**

-	Problem formulation: first you use x_n as index (l. 96), later x_t (eq. 1, l. 105) to denote the sampling points -> is this on purpose for spatial and temporal resolution or should it always be x_t? Please check carefully.
-	L. 108: covariance between z and sampled points -> what is z? I assume you refer to the latent space but this is not clear here, please clarify.
-	Algorithm 1: line 13, please check type setting again, arrow is weird
-	L. 227: what is the observation space? (num of muscles + num of joints?)
-	L. 266 should be semitendinosus (s missing)

**Robotics Focus:**

3

**Summary Of Paper:**

This paper presents a novel approach (HDSAFEBO) to safely optimize the control of high-dimensional systems at the example of musculoskeletal systems. The results indicate that the HDSAFEBO outperforms existing approaches in terms of efficiency while still having a high safety probability. Two examples are showcased: the control of an upper limb (55 muscles) rotating a bottle in simulation and a neural stimulation of a lower limb (both in simulation and with a paraplegic patient).

**Summary Of Recommendation:**

This paper is technically sound, and the question of safe optimization in high-dimensions addresses a core robotics challenge. However, the comparison focuses almost solely on BO methods and neglecs other high-dimensional optimization methods.

---

### Official Review · Reviewer_KPXx · 2024-07-20

**Originality:** 2
**Technical Quality:** 2
**Clarity Of Presentation:** 2
**Potential Impact:** 2
**Recommendation:** 2
**Confidence:** 4

**Review:**

Strengths:
- The paper tackles the important problem of high-dimensional safety-critical control.

Concerns:
- It’s not 100% clear what the paper’s technical contribution is over existing methods. For instance, several papers in the related work (like [36-41]) also use dimensionality reduction techniques, e.g., autoencoders, and then run BO in the latent space. To my understanding, the primary novelty is in using optimism in sample selection (to provide some safety), and in integrating local search. Thus, testing two ablations where HDSafeBO is run without the optimistic sample selection/the local search procedure would help make the contribution more clear.
- In all practical situations, autoencoder training is not exact, e.g., encoding and decoding a high-dimensional sample does not exactly return the original sample. As such, \Pi^{-1} is not exact, and thus to my understanding Proposition 4.3 does not hold for this practical setting. It should be made clear in the text, or in the limitations section, that the safety guarantees do not hold in practical situations where the decoder is not trained to be a perfect inverse of the encoder.
- How well upper-confidence sample selection works seems entirely dependent on the size/shape of the feasible set of the specific problem. One can imagine that for highly-constrained problems, upper confidence would lead to frequent safety violations. Some discussion in the paper with regard to when upper-confidence sample selection can be expected to work would be helpful.
- Line 156-158: the chosen 0.5 threshold is not motivated. Why is 0.5 the maximum acceptable level of violation?
- Proposition 4.1 reasons about how each individual point is safe with some probability, and then a guarantee on the cumulative safety violation is constructed using the individual safety violation probabilities per trial using I believe Boole’s inequality. However, wouldn’t one want to directly compute the joint probability of the T trials violating the safety constraints, as this would be less conservative? This is done, for instance, in [A].
- Line 134-135: “the discrete nature of our safety estimation and search space” -> this is unclear to me - aren’t the experiments run for continuous control problems? For instance, Sec. 5.2 is optimizing over continuous parameters in a linear control policy (line 226).
- Why is Assumption 3.1 not also required for the function f?
- \Pi is never defined formally in the text (only mentioned in the algorithm block)

[A] Chou, Wang, Berenson. Gaussian Process Constraint Learning for Scalable Chance-Constrained Motion Planning from Demonstrations. RA-L 2022.

**Quality Of The Limitations Section:**

2

**Questions For Rebuttal:**

Please see review

**Robotics Focus:**

4

**Summary Of Paper:**

This paper proposes a method for safe high-dimensional control through Bayesian optimization (BO). The typical scalability problems of BO are mitigated through various dimensionality reduction techniques. Optimistic safety identification and local search are used to improve performance. The method is evaluated on two high-dimensional control tasks, outperforming baselines.

**Summary Of Recommendation:**

I currently recommend a weak reject due to a number of unclear points and claims in the writing, and an unclear technical contribution. I'm open to raising my score if the concerns raised in the review are addressed.

---

### Official Review · Reviewer_7rTb · 2024-07-27
**A good submission (although I cannot judge novelty)**

**Originality:** 3
**Technical Quality:** 4
**Clarity Of Presentation:** 5
**Potential Impact:** 3
**Recommendation:** 4
**Confidence:** 3

**Review:**

Strengths:

The paper is well written and easy to follow (with a working knowledge of Bayesian optimization).
The algorithmic ingredients are simple yet seem to work effectively in combination.
Although no robot experiments are presented, the authors performed a real-world study on a paraplegic patient.
For all experiments, the authors provide either explanations of the objective functions in the appendix and/or provide videos of the optimised behaviour.
I appreciate the connection between the trust-region size and the maximum information gain which in turn influences the cumulative safety violation. Although no formal proof, this argument helps understanding the safer exploration of HDSafeBO.

Weaknesses:

I do not see major weaknesses in general with the paper. The method seems to be sound and working.

Apart from experimental improvements which I'll mention next, I simply want to state that I am not an expert in the related literature of this domain and hence cannot judge whether similar approaches have already been developed. I am only mentioning this because the individual algorithmic ingredients are relatively simple which is why I would be surprised if nobody else had tried this before. If indeed the authors are the first to try this out, I congratulate them for creating this simple yet effective recipe for BO.

In terms of experiments, I would have the following suggestions for improvement:
 * It would be interesting to see the baseline method performances in Section 5.1 and 5.3 if they were granted the same embedding as HDSafeBO. In Section 5.2, the authors already granted access to the embedding since no baseline method could otherwise deal with the search space otherwise.
 * Given the theoretical impact of the trust-region on the safety of the algorithm, I would be highly interested in an ablation of the method without the trust-region.
 * Furthermore, I would appreciate an experiment that investigates the algorithm's performance w.r.t. to the quality of the isometric embedding. While the authors note that they did not observe problems with imperfect embeddings, it would be interesting to see just how imperfect they can be.

Comments:

 * Is the reward function for the musculoskeletal model control in the appendix wrong? It seems like the signs are flipped since dropping the bottle leads to a large positive reward.
 * In Algorithm 2, shouldn't it be $\min(2 l_{t-1}, l_{max})$ in line 7 and $max(0.5 l_{t-1}, l_{min})$ in line 9?

**Quality Of The Limitations Section:**

3

**Questions For Rebuttal:**

Sea weaknesses and comments section above

**Robotics Focus:**

3

**Summary Of Paper:**

The paper presents a Bayesian optimisation algorithm that targets optimisation in high-dimensional spaces while providing probabilistic safety guarantees. The authors use a combination of isometric dimensionality reduction, trust-region approaches, and confidence bounds to realise the algorithm. The proposed combination of these techniques consistently outperforms alternative methods in experiments.

**Summary Of Recommendation:**

I think the paper should be accepted at the conference if there is no significant algorithmic overlap with prior work.

---

### Author Rebuttal · Authors · 2024-08-09

We would like to thank the AC and the reviewers for the evaluations. We believe that our paper has benefited from their helpful comments. We have revised the manuscript accordingly and are committed to making further improvements based on the suggestions provided. Please find our revised version and additional experimental results attached in the rebuttal ZIP file below. We will also organize the paper structure and include the additional experimental results from the rebuttal in the camera-ready version.

---

### Decision · Program_Chairs · 2024-09-04

**Decision:**

Accept

**Comment:**

In general, the reviewers are positive about this paper. Most reviewers like the simplicity and effectiveness of the method.
There are still some points to clarify. Specifically, the authors should try to highlight the novelty of the approach and frame it better in the literature. On top of that, I would strongly advise the authors to add more BO baselines to the comparison.
Another minor issue is the proper handling of the limitation section, which should make clear when the safety properties hold and how the safety problem is relaxed, particularly when the encoder is not perfectly trained.

=====

After the rebuttal, most of the reviewers were convinced by the author's response. The additional ablations to assess the impact of algorithmic design choices were particularly appreciated.
Please, make sure to open-source your code to enhance the reproducibility of this work, as discussed with the reviewers.